# Experiment and Modeling on Macro Fiber Composite Stress-Induced Actuation Function Degradation

**Wei Wang** *, **Zikuo Zhang and Zhichun Yang**

Department of Aeronautical Structural Engineering, School of Aeronautics, Northwestern Polytechnical University, Xi'an 710072, China; zhangzkuo@163.com (Z.Z.); yangzc@nwpu.edu.cn (Z.Y.)
* Correspondence: wwang@nwpu.edu.cn

**Abstract:** The effect of stress depolarization will cause actuation function degradation of a piezoelectric actuator, which can eventually trigger function failure of the piezoelectric smart structure system. In the present study, we experimentally demonstrate the degradation process of the actuation function of the Macro Fiber Composite (MFC) piezoelectric actuator. Actuation function degradation data of MFC actuators undergoing cyclic loads with four different stress amplitudes have been measured. Based upon the experimental results, the radial basis function (RBF) neural network learning algorithm was adopted to establish a neural network model, in order to predict the actuation function degenerative degree of the MFC actuator, undergoing arbitrary cyclic load within the concerned stress amplitude range. The maximum relative error between the predicted result and our experimental result is 4%.

**Keywords:** stress induced; MFC actuator; actuation function degradation; experimental observation; predictive modeling

---

## 1. Introduction

The functional structural system of piezoelectric smart/intelligent structure was developed rapidly. A large amount of research works are devoted to the application study of piezoelectric elements, such as the piezoelectric actuator, sensor and energy transducer in structural vibration control, structure health monitoring and energy harvesting. Nevertheless, performance reliability of the three mentioned types of piezoelectric elements is very important for the practical application of the piezoelectric smart structure system. Experimental results of piezoelectric ceramic samples show remarkable degradation of piezoelectric properties after mechanical loading due to the stress depolarization effect [1–9]. In applications, the piezoelectric properties' degradation of piezoelectric ceramics in piezoelectric actuators is inevitable, due to the effect of stress depolarization when piezoelectric actuators and the main structure bear the external load simultaneously. Such an occurrence will degrade the actuation function of the piezoelectric actuator and finally lead to the function failure of the piezoelectric smart structure system.

Recently, function degradation of piezoelectric transducers in an energy harvesting system was studied by a few researchers. Pillatsch et al. [10] studied the degradation of the energy conversion function of piezoelectric ceramics patch in the piezoelectric energy harvesting system. Bimorph piezoelectric beams were subjected to lifespan testing through electromagnetic tip actuation with a large number of cycles. Experimental results show that the degradation of the piezoelectric material will lead to the resonance frequency shift of the double crystal piezoelectric beam, which has a strong impact upon the efficiency of the energy collecting system utilizing resonance mode for energy harvesting.

Deepesh Upadrashta and Yaowen Yang [11] studied the functional reliability of Macro Fiber Composite (MFC) in energy harvesting system. Aimed to obtain the upper limit of dynamic strain

on MFC, this strain limit is expected to be used as a failure limit in the design of piezoelectric energy harvesters (PEHs). Based on experimental results, 600 $\mu\varepsilon$ was proposed to be the upper limit of safe strain amplitude for a reliable performance of MFC in the energy harvesting system. P. Pillatsch et al. [12] investigated the degradation of bimorph piezoelectric bending beams under symmetrical and asymmetrical sinusoidal loading. It is concluded that the degradation of the bimorph piezoelectric patch will bring a loss in output power and the shift in resonance frequencies, and micro-cracking was shown to occur predominantly in piezoelectric layers under tensile stress. James Kuria Kimotho, Tobias Hemsel, and Walter Sextro [13] proposed a prognostic approach which involves a training machine learning algorithm to model the degradation of the piezoelectric transducers through a health index, and a use of the learned model to estimate the health index of similar transducers, where electric signals of the piezoelectric transducer are used as the condition monitoring data. Mahesh Peddigari et al. [14] investigated the reliable performance of hard and soft piezoelectric single crystal-based fiber composites (SFC)-based piezoelectric energy harvesters (PEHs). Base acceleration of both PEHs was held at 7 m/s$^2$ and the frequency of excitation is tuned to their resonant frequency, and then the output power was monitored for $10^7$ fatigue cycles. Experimental results indicate that fatigue-induced performance degradation in soft SFC-based PEH is more prominent than that in hard SFC-based PEH.

The energy conversion efficiency of the piezoelectric transducer is the highest when it works in its resonance state. Above-mentioned researches have been mainly focused on the function degradation test of a piezoelectric transducer under this resonance condition. On the other hand, the piezoelectric actuator is the key component of a piezoelectric active control system for structural vibration control. Actuation function degradation of the piezoelectric actuator will directly cause the function failure of the piezoelectric active control system. Little attention has been paid to the actuation function degradation of the piezoelectric actuator. In the present study, actuation function degradation of the piezoelectric actuator undergoing cyclic load is our concern.

MFC actuators possess good electromechanical coupling performance and excellent structure adaptability. It has great potential in the application of a piezoelectric smart control system for aircraft structural vibration active control, such as aircraft tail buffeting control and wing flutter suppression. The present paper investigates the actuation function degenerative behavior of the MFC actuator and the actuation function degradation prediction of the MFC actuator undergoing cyclic load. Actuation function degradation data of MFC actuators undergoing cyclic loads with four different stress amplitudes have been measured. Based on the experimental results, the radial basis function (RBF) neural network learning algorithm is adopted to establish the prediction model, which can be used to predict the actuation function degradation degree of the MFC actuator undergoing the arbitrary stress amplitude of the cyclic load within the concerned stress amplitude range.

## 2. Experiment Set Up and Procedures

### 2.1. Actuation Function Degradation Characterization of Macro Fiber Composite (MFC) Actuator Undergoing Cyclic Load

In order to quantitatively characterize the actuation function degradation behavior of an MFC actuator undergoing cyclic load, an experimental test procedure is established to measure the actuation function degradation degree of the MFC actuator at different cyclic loads.

The MFC actuator is a type of flexible patch actuator, and it is not convenient to directly apply cyclic load on the actuator to implement the measurement of its actuation function degradation. In order to carry out the measurement of the MFC acutator undergoing cyclic load, this MFC actuator needs to be made into an MFC piezoelectric beam specimen. Resonant displacement amplitude of the MFC piezoelectric beam actuated by the attached MFC actuator will be measured and used to characterize the actuation function of the MFC actuator.

The relative decrease percentage of measured resonant displacement amplitudes will be used to indicate the actuation function degradation degree of the MFC actuator. Degradation degree versus the cycles of cyclic load will provide a group of degradation data. The fitting curve of these data can be

used to exhibit the actuation function degradation tendency of the MFC actuator undergoing cyclic load. To get these relative decrease percentages of displacement amplitudes, the resonant amplitude of an MFC piezoelectric beam actuated by the intact MFC actuator needs to be measured first of all, and this value is then denoted as $A_0$. A shaker is used to impose cyclic load on the MFC actuator. When the cycles of cyclic load $N$ reaches a certain number, the cyclic load is paused for an actuation function test. The resonant displacement amplitude in the free end of the beam actuated by MFC is measured under the same driving condition as for the intact MFC, which is denoted by $A_t$ $(\sigma, N)$, in which $\sigma$ is the stress amplitude of the cyclic load and $N$ represents the cyclic loading cycles. The actuation function degradation degree of our MFC actuator can be characterized by the actuation function index $A$ $(\sigma, N)$, which is defined as

$$A(\sigma, N) = \frac{A_t(\sigma, N)}{A_0} \times 100\% \tag{1}$$

where $A_0$ is the measured resonant displacement amplitude in the free end of the MFC piezoelectric beam actuated by the intact MFC actuator. $A_t$ $(\sigma, N)$ is the free end resonant displacement amplitude of the MFC piezoelectric beam actuated by MFC actuator, which can be measured right after $N$ cycles of cyclic load with stress amplitude $\sigma$ has been imposed on the MFC actuator.

For an intact MFC actuator the actuation function index $A$ (0,0) = 100%. When $A$% is less than 100% it means that the actuation function degradation of the MFC actuator occurs. In other words, the actuation function of the MFC actuator degenerates to $A$% of its intact level.

In the calculation of this actuation function index $A$ $(\sigma, N)$, it should be ensured that the used two resonant displacement amplitudes are comparable, which means the corresponding driving conditions and boundary conditions of the beam for the measurement of the two resonant displacement amplitudes must be the same. Hence for the measurements of the two resonant amplitudes $A_t$ $(\sigma, N)$ and $A_0$, the amplitude and the frequency of the driving voltage for the MFC actuator both need to remain consistent, where the frequency should be confirmed as the first bending mode natural frequency of the MFC piezoelectric beam under the corresponding boundary conditions.

*2.2. Actuation Function Degradation Test of MFC Actuator Undergoing Cyclic Load*

In the present test, the MFC actuator is bonded to the middle surface of a steel beam to form an MFC piezoelectric beam test specimen. Cyclic loads are imposed on the MFC actuator through the bending resonant vibration of the beam excited by a shaker. Actuation function degradation data and curves of MFC actuation function index $A$ versus $N$-cycles of cyclic load have been obtained for cyclic loads with four different stress amplitudes.

The actuation function of an intact MFC actuator is characterized by the amplitude $A_0$, which is the free end resonant displacement amplitude of the MFC piezoelectric beam actuated by the intact MFC actuator under certain input driving conditions for MFC. Then the MFC actuator is cyclically loaded by a shaker via the beam specimen. For every 50,000 cycles of cyclic load, there sonant amplitude $A_t$ $(\sigma, \mathrm{N})$ in the free end of the beam which is actuated by the attached MFC actuator is measured under the same input driving condition as that of $A_0$. According to Equation (1) the actuation function index $A$ $(\sigma, N)$ of the MFC actuator can be obtained, and it can be used to characterize the degenerate degree of the MFC piezoelectric actuation function undergoing $N$ cycles of cyclic load with stress amplitude $\sigma$. The test procedure of the MFC actuator actuation function degradation undergoing cyclic load is shown in Figure 1.

According to the engineering properties of the MFC actuator, the linear-elastic tensile strain limit is 1000 $\mu\varepsilon$ [14], and the corresponding stress limit is 30 MPa. Considering this limit, cyclic loads with four different stress amplitudes are implemented for the actuation function degradation test of our MFC actuator. Four stress amplitudes of the cyclic loads are 12 MPa, 18 MPa, 24 MPa and 30 MPa. The corresponding dynamic strain amplitudes of the MFC actuator for these four stress amplitudes are:400 $\mu\varepsilon$, 600 $\mu\varepsilon$, 800 $\mu\varepsilon$ and 1000 $\mu\varepsilon$, respectively.

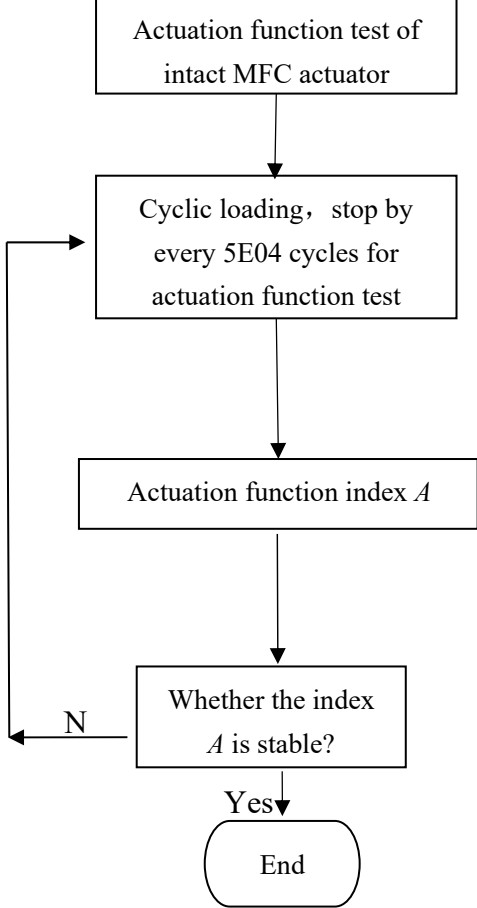

**Figure 1.** Flow diagram for the actuation function degradation test procedure of the Macro Fiber Composite (MFC) actuator.

The experimental results of the sensing function degradation test for the MFC sensor show that, when the MFC sensing function degradation degree is greater than 10%, which can be characterized by the reduction of its output voltage, it is considered that its function degradation is observable for the corresponding test case [12]. Hence, in the present test when the measured index *A* is greater than 90%, which indicates that the actuation function degradation degree is less than 10%, it will be considered that there is no actuation function degradation for the corresponding load cycles. The termination number of loading cycles for the present test is determined according to the variation of index *A*. When the index *A* is basically stable, which indicates no observable further actuation function degradation can be confirmed, this function degradation test will be terminated.

In the present study, an M2814-P1 MFC actuator is selected as our test object. Properties of this MFC actuator can be found in Table 1. Four of the same type of MFC actuators have been made into four MFC piezoelectric beam specimens to implement the actuation function degradation tests, which are shown in Figure 2. The four MFC piezoelectric beam specimens will be used for cyclic loading and actuation function degradation tests of cyclic loads with stress amplitudes 12 MPa, 18 MPa, 24 MPa and 30 MPa, respectively.

The test layout of the cyclic loading experiment for the MFC actuator is shown in Figure 3. In the cyclic loading experiment of the MFC actuator, the MFC piezoelectric beam is fixed by one end, and the resonant bending vibration of the MFC piezoelectric beam is excited by a shaker, whose action point is near to the fixed end of the beam. The excitation signal input to the power amplifier of the shaker is a sinusoidal signal, and the frequency is equal to the first natural mode (first bending mode) frequency of the MFC piezoelectric beam specimen. By adjusting the input excitation voltage amplitude, the excitation force of this shaker can be adjusted to control the stress (strain) amplitude of

the cyclic loads imposed upon the MFC actuator. Cyclic loads with different stress amplitudes can be applied to the MFC actuator via the beam specimen under the excitation of a shaker, and the stress (strain) amplitude of the cyclic load imposed on the MFC actuator is measured by a foil strain gauge, which has been bonded onto the surface of MFC actuator. A dynamic strain gauge is used to measure the dynamic strain of the MFC actuator under the action of cyclic loads, and the stress amplitude of the cyclic load will be measured and recorded by the strain gauge during the whole process of the cyclic loading test. In the present study only the cyclic load is imposed on the MFC actuator without an external electric field.

**Table 1.** Properties of the MFC actuator [15].

| MFC (M2814-P1) | |
| --- | --- |
| Full size (mm) | $60 \times 32 \times 0.3$ |
| Active size (mm) | $56 \times 28 \times 0.3$ |
| Fiber Material | PZT-5A |
| $d_{33}$ (pm/V) | 400 |
| $E_1$ (GPa) | 30.34 |
| E2 (GPa) | 15.85 |

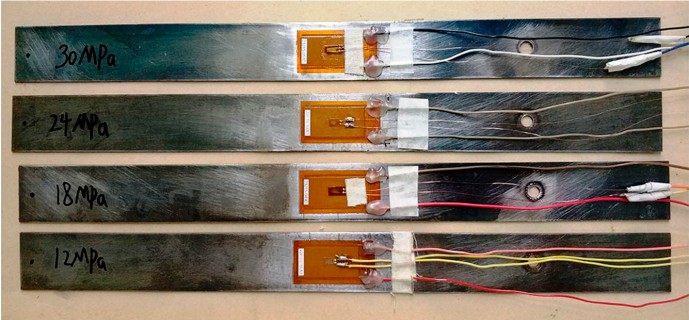

**Figure 2.** Test specimens of a steel beam bonded with the MFC (M2814-P1) actuator, and a foil strain gauge bonded on the surface of the MFC actuator.

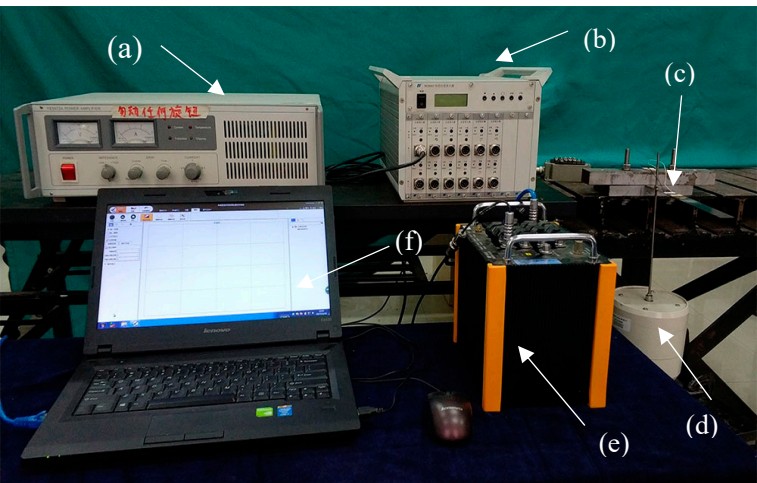

**Figure 3.** Photograph of the cyclic loading system for the MFC actuator. (**a**) Amplifier of shaker; (**b**) Dynamic strain gauge; (**c**) Steel beam with MFC; (**d**) Shaker; (**e**) Signal Acquisition System (**f**) PC.

During the experiment, cyclic loading will be paused by every 50,000 cycles for an actuation function degradation test. The actuation function index *A* of the MFC actuator will be measured immediately, and then cyclic loading will be continued for another 50,000 cycles. For the actuation function test of the MFC actuator, a laser displacement sensor is used to measure the free end resonant



displacement response time history of the MFC piezoelectric beam actuated by the attached MFC actuator, and the resonant amplitude $A_t$ $(\sigma, N)$ will be obtained for this specific stress amplitude and cycles of the cyclic load. The experiment set up for actuation function test system of the MFC actuator is shown in Figure 4.

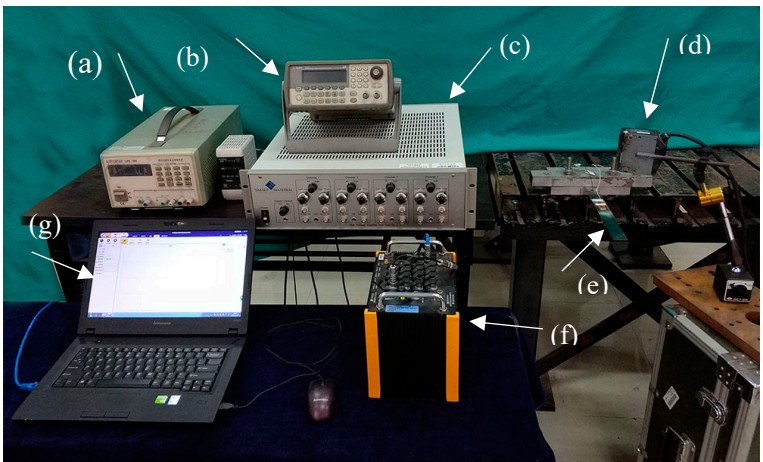

**Figure 4.** Actuation function test set up for the MFC actuator: (**a**) DC power; (**b**) Signal generator; (**c**) High voltage amplifier; (**d**) Laser sensor; (**e**) Steel beam with MFC; (**f**) Signal acquisition system (**g**) PC.

For the measurement of the two amplitudes $A_0$ and $A_t$, which are used to calculate the actuation function index $A$, the amplitude and frequency of the driving voltage for the MFC actuator should be ensured to be consistent. In the present test, the frequency of the driving voltage for MFC is 13.67 Hz, which is the first natural mode frequency of the MFC piezoelectric beam under the corresponding boundary condition. Three different amplitudes of driving voltage are chosen as 160 V, 200 V and 240 V for the actuation function degradation test of MFC actuator.

## 3. Results and Discussion

Free end resonant displacement response time history graphs of the MFC piezoelectric beams actuated by MFC actuators are shown in Figures 5–7. Reduction of the displacement amplitude can be clearly observed after a certain number of cyclic loads has been imposed on the MFC actuator. These test results evidently show the actuation function degradation of the MFC actuator undergoing cyclic loads with stress amplitudes of 18 MPa, 24 MPa and 30 MPa, respectively.

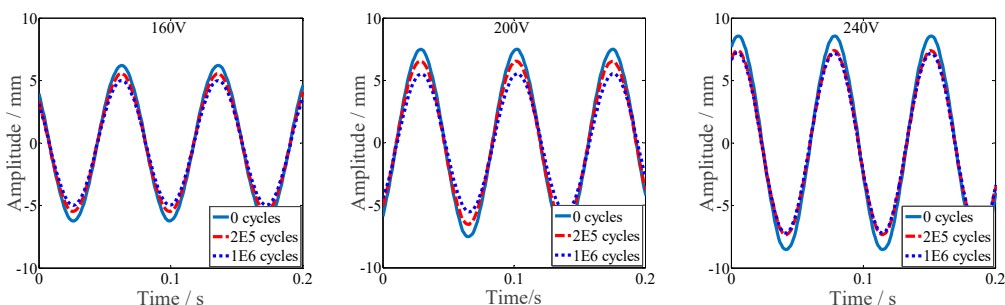

**Figure 5.** Tip displacement time history of the beam specimen actuated by MFC under cyclic load (stress amplitude 18 MPa).

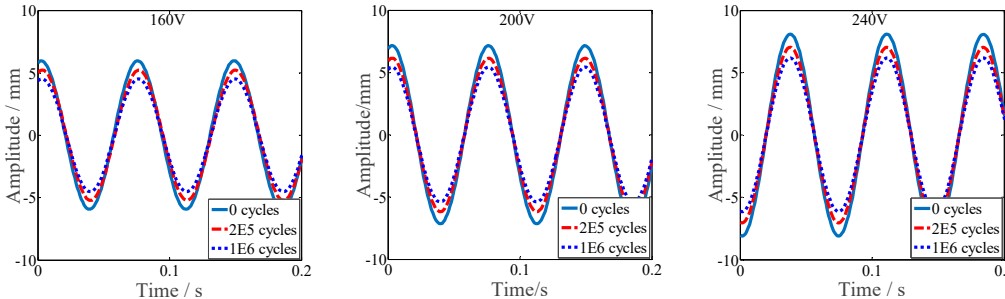

**Figure 6.** Tip displacement time history of the beam specimen actuated by MFC under cyclic load (stress amplitude 24 MPa).

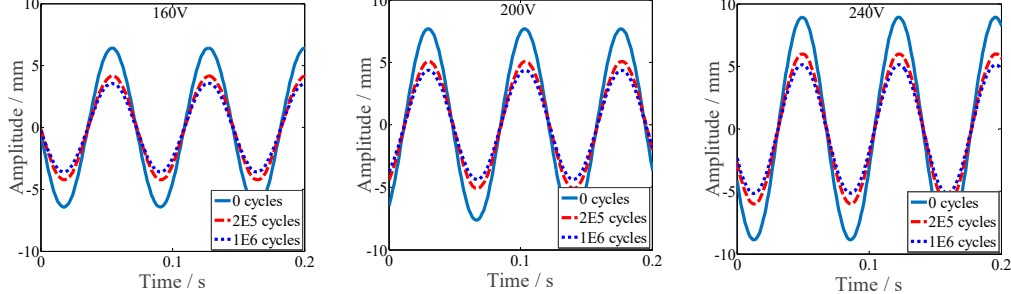

**Figure 7.** Tip displacement time history of the beam specimen actuated by MFC under cyclic load (stress amplitude 30 MPa).

The resonant displacement response amplitudes in the free end of the beam actuated by the MFC actuator undergoing different amplitudes of cyclic load versus the loading cycles are shown in Figure 8. As it can be seen from Figure 8a, when the stress amplitude of the cyclic load is 12 MPa, the resonant displacement amplitude of the piezoelectric cantilever beam actuated by MFC does not degenerate with the increase of load cycles. It indicates that the actuation function of the MFC actuator does not degenerate undergoing cyclic loads whose stress amplitude is less than 12 MPa.

The experimental results in Figure 8b–d show that for the cyclic loads whose stress amplitudes are 18 MPa, 24 MPa and 30 MPa, respectively, degradation of the free end resonant displacement amplitude of the beam specimen actuated by the MFC actuator can be visibly observed with the increase of the loading cycles. These results evidently indicate that the actuation function of the MFC actuator does degenerate with the increase of loading cycles for some stress amplitude level. It can also be seen, from the slope change of the amplitude degenerate curves, that the actuation function degradation rate increases significantly with the increase of the stress amplitude.

By using the calculation formula of the actuation function index $A(\sigma, N)$, the variation curves of the MFC piezoelectric actuation function index $A(\sigma, N)$ versus loading cycles have been plotted for cyclic loading stress amplitudes of 30 MPa, 24 MPa, 18 MPa and 12 MPa, respectively, which can be seen in Figure 9.

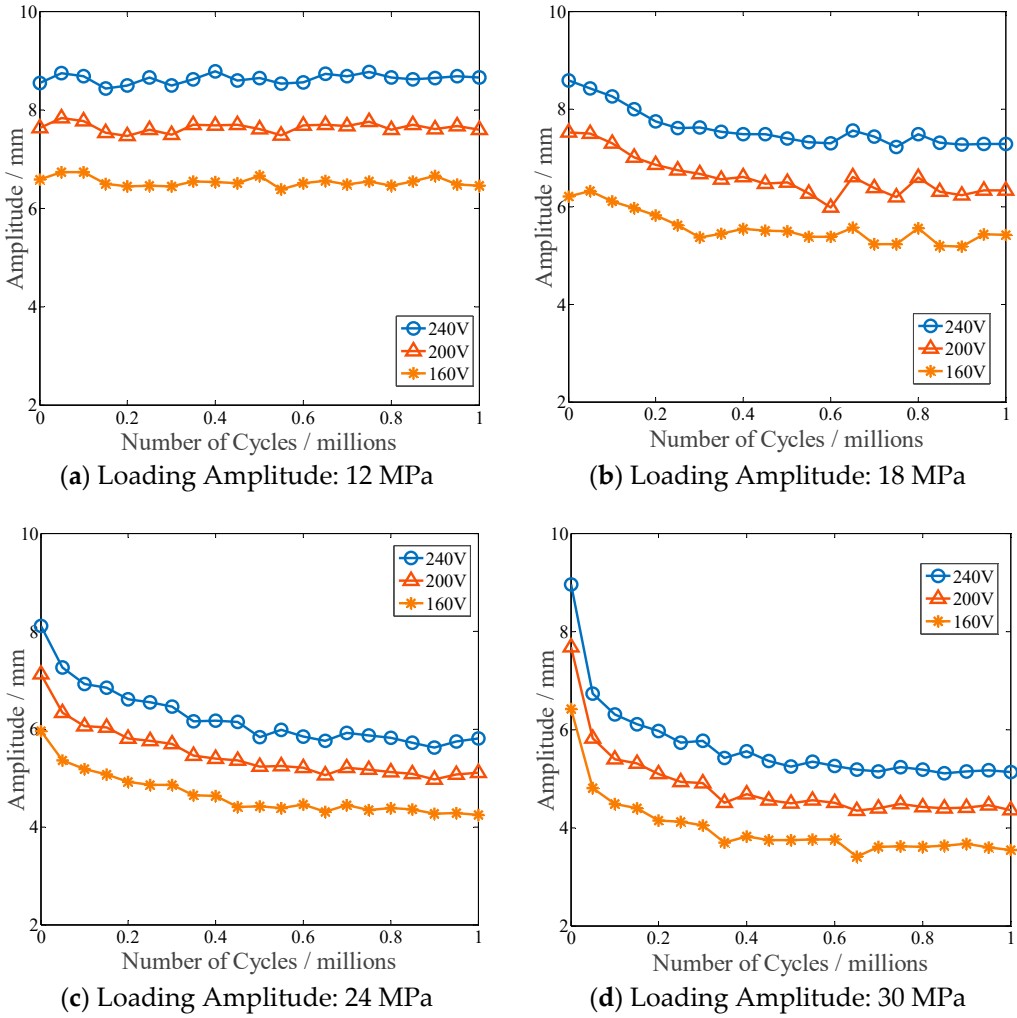

(**a**) Loading Amplitude: 12 MPa

(**b**) Loading Amplitude: 18 MPa

(**c**) Loading Amplitude: 24 MPa

(**d**) Loading Amplitude: 30 MPa

**Figure 8.** MFC actuated displacement amplitude of the four tested beams versus load cycles.

As can be seen from the variation curves of the actuation function index $A(\sigma, N)$ shown in Figure 9, for the cyclic loads with stress amplitudes 30 MPa, 24 MPa and 18 MPa, the MFC piezoelectric actuation function gradually degenerates with the increase of the cyclic load cycles, and the degree of actuation function degradation is independent from the amplitude of the excitation voltage applied to the MFC actuator. As it can be seen from Figure 9a that the actuation function of MFC degenerates rapidly within the first $4 \times 10^5$ cycles of cyclic load with stress amplitude 30 MPa. For example, when the number of cycles only reaches the first 250,000, the MFC piezoelectric actuation function has already degenerated to 66% of its intact level.

The fitting curve of the MFC actuation function index $A(\sigma, N)$ versus load cycles demonstrates that the actuation function degradation process of the MFC actuator undergoing cyclic load is a nonlinear process. From the slope change of the actuation function degradation curve, it can be found that, within the first $4 \times 10^5$ cycles, the actuation function of MFC degrades rapidly, then after $8 \times 10^5$ cycles the degenerate rate gradually slows down with the increase of the loading cycles. The actuation function of MFC undergoing cyclic loading with stress amplitudes of 30 MPa, 24 MPa and 18 MPa degenerates by 43%, 29% and 15%, respectively, when the cycles of cyclic load reach $1 \times 10^6$.

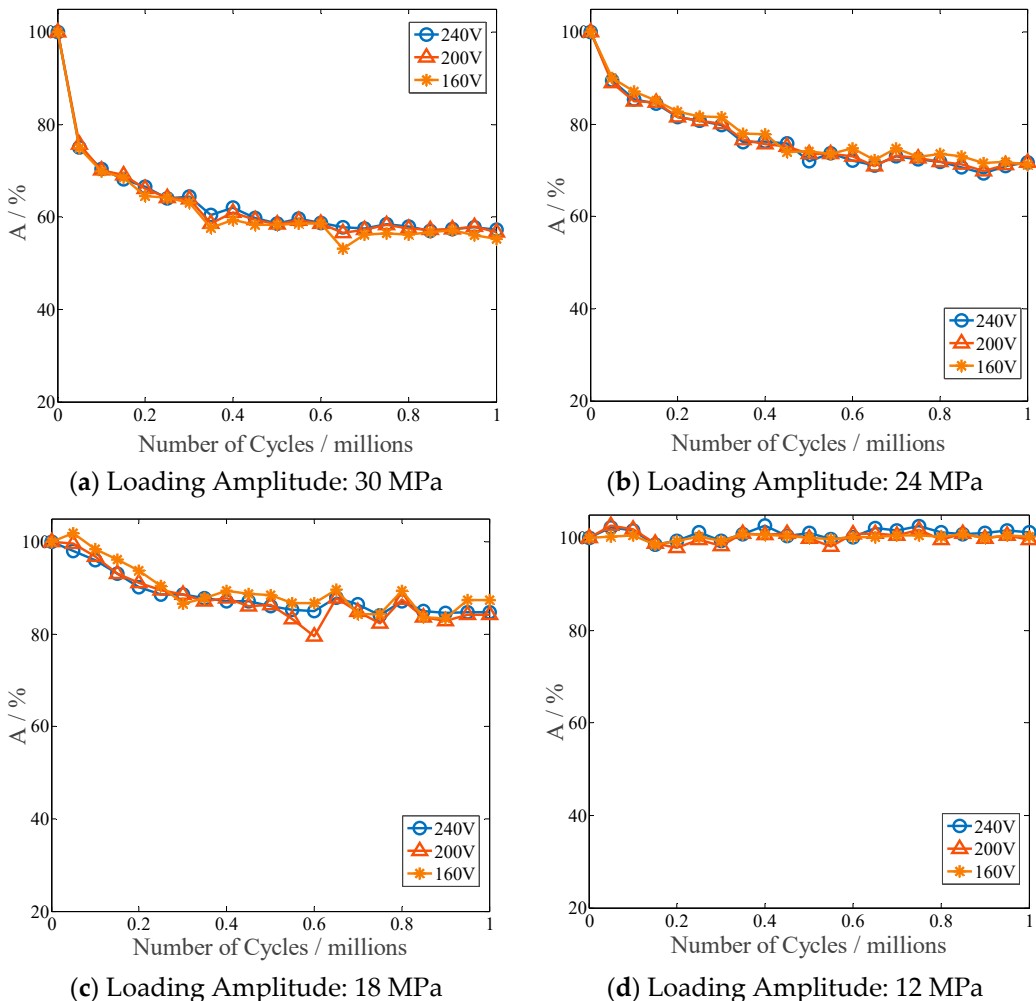

**Figure 9.** Actuating function index *A* of MFC actuator versus loading cycles.

## 4. Predictive Modeling of Stress Induced MFC Actuation Function Degradation

The degradation of material performance is always accompanied by some form of material nonlinear mechanical behavior, hence the macroscopic appearance of degradation will also be nonlinear. The radial basis function (RBF) neural network model can approach any non-linear functions [16], and this feature is used for modeling the degradation paths based on known degradation data. In order to obtain a prediction model for the actuation function degenerate degree prediction of the MFC actuator undergoing cyclic load, the RBF neural network learning algorithm is adopted. The established actuation function degradation prediction model of the MFC actuator can be used to predict the actuation function index *A* of the MFC actuator for arbitrary given stress amplitude and cycles of cyclic load within concerned stress amplitude range.

The RBF radial basis function neural network is a forward network composed of three layers [17]. The first layer is the input layer composed of input nodes. The second layer is hidden layer, the neurons number of which needs to be determined. The third layer is the output layer, it is the response of the input layer, and the number of the output nodes is determined according to the dimensions of our output data. The key to establish this RBF neural network model is to determine the distributed parameters spread and the number of hidden layer nodes *mn*. The larger the spread is, the smoother the fitted curve will be, but the fitting accuracy will be reduced. The smaller the spread is, the higher the curve fitting accuracy will be, but the prediction ability of new data will get lower. The number of neurons in the hidden layer, *mn*, is not the more the better. Simply pursuing the fitting accuracy, too many neurons will also lead to over fitting. Generally the number of neurons in the hidden layer

can be determined according to the empirical formula $mn = \sqrt{mn_p + mn_T} + n$, in which $mn$, $mn_p$, $mn_T$ are the numbers of neurons in the hidden layer, input layer and output layer, respectively, and $n$ is any integer between 1 and 10.

The main procedure of establishing the prediction model using the RBF neural network learning algorithm is summarized as follows: Firstly, normalize the input and output, which are the cycles of the given stress amplitude cyclic load and the corresponding measured actuation function index $A$ $(\sigma, N)$, and then use these data to train the neural network, and calculate the square sum of the fitting error, the square sum of the prediction error and the sum of the total error. The distribution parameter of the neural network and the number of neurons in the hidden layer are adjusted to get the highest fitting accuracy for describing the actuation functional degenerate curve. Finally, the prediction model will be established with the settled parameters.

In order to demonstrate the feasibility of the above-mentioned procedure, the measured data of 12 MPa, 18 MPa and 30 MPa are used as the training samples to train the prediction model of the RBF neural network. The modeling process was implemented in MATLAB by using the "newrb" function. When the number of neurons in the hidden layer, $mn = 25$, and the distribution parameter of neural network spread = 0.5731, the neural network model has the highest accuracy for describing the measured actuation function degradation of the MFC actuator. Table 2 shows the model parameters of the RBF neural network model for the actuation function degradation prediction of the MFC actuator.

**Table 2.** Parameters of radial basis function (RBF) neural network model for actuation function degradation prediction.

| Neuron Number in Hidden Layer | Radial Basis Function Center | | Weight between Hidden Layer and Output Layer |
|---|---|---|---|
| 1 | −0.10 | −1.00 | $4.51 \times 10^{-1}$ |
| 2 | 0.30 | 1.00 | $1.75 \times 10^{4}$ |
| 3 | 1.00 | −1.00 | $5.48 \times 10^{-1}$ |
| 4 | −1.00 | −1.00 | $2.38 \times 10^{-1}$ |
| 5 | −1.00 | −0.33 | $-5.35 \times 10^{2}$ |
| 6 | 1.00 | 1.00 | $-5.63 \times 10^{3}$ |
| 7 | 0.00 | 1.00 | $-9.81 \times 10^{4}$ |
| 8 | −0.90 | −0.33 | $1.73 \times 10^{3}$ |
| 9 | −1.00 | 1.00 | $6.33 \times 10^{4}$ |
| 10 | −0.90 | 1.00 | $1.87 \times 10^{4}$ |
| 11 | −0.80 | 1.00 | $-1.52 \times 10^{6}$ |
| 12 | −0.70 | 1.00 | $5.64 \times 10^{6}$ |
| 13 | 1.00 | −0.33 | $3.19 \times 10^{-1}$ |
| 14 | −0.60 | 1.00 | $-1.03 \times 10^{7}$ |
| 15 | −0.50 | 1.00 | $1.11 \times 10^{7}$ |
| 16 | 0.40 | −1.00 | $7.53 \times 10^{-2}$ |
| 17 | −0.40 | 1.00 | $-6.90 \times 10^{6}$ |
| 18 | −0.30 | 1.00 | $2.03 \times 10^{6}$ |
| 19 | 0.90 | 1.00 | $1.68 \times 10^{4}$ |
| 20 | 0.80 | 1.00 | $-1.39 \times 10^{4}$ |
| 21 | −0.80 | −0.33 | $-2.01 \times 10^{3}$ |
| 22 | −0.30 | −0.33 | $-5.02 \times 10^{1}$ |
| 23 | 0.50 | −0.33 | $-0.36 \times 10^{1}$ |
| 24 | −0.70 | −0.33 | $8.55 \times 10^{2}$ |
| 25 | 0.10 | −0.33 | $1.25 \times 10^{1}$ |

The established model has been used to predict the degradation curve for the actuation function degradation of the MFC actuator undergoing the cyclic load with the stress amplitude of 24 MPa. Firstly, the fitting curves for the three groups of measured MFC piezoelectric actuation degradation data are built, and the corresponding degenerate curves can be obtained, as shown with the dashed

line in Figure 10. The fitting results of the three sets of measured data are shown in the second blocks of Table 3. The corresponding test results are given in the first column block of Table 3. Then, the learning algorithm of the RBF neural network is used to establish the prediction model by these fitting curves. Finally, the built prediction model is used to predict the MFC piezoelectric actuation function degradation curve undergoing cyclic load with stress amplitude of 24 MPa, which is shown with the red solid line in Figure 10.

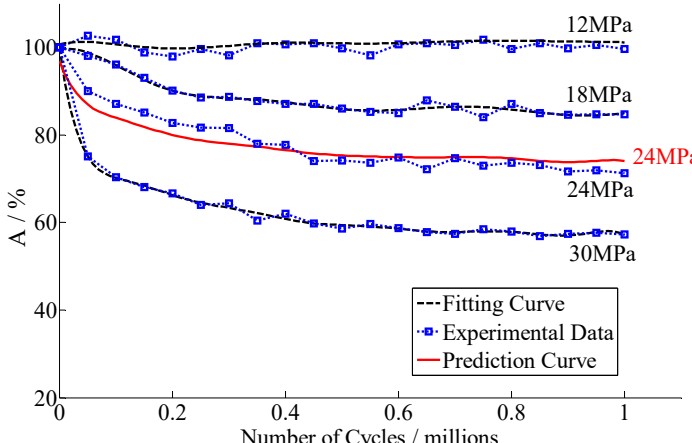

**Figure 10.** Predicted actuation function degradation curve (red solid line) of the MFC actuator undergoing cyclic load with stress amplitude of 24 MPa.

**Table 3.** Actuation function degradation prediction and comparison.

| | Test *A*% | | | Fitting *A*% | | | Prediction *A*% | | | Cycles |
|---|---|---|---|---|---|---|---|---|---|---|
| | | | | | | | | 24 MPa | | |
| 30 MPa | 18 MPa | 12 MPa | 30 MPa | 18 MPa | 12 MPa | Test | Predict | Error | |
| 100 | 100 | 100 | 100 | 100 | 100 | 100 | 97 | 3% | 0 |
| 75 | 98 | 100 | 75 | 99 | 100 | 90 | 87 | 3% | $5.0 \times 10^4$ |
| 70 | 96 | 100 | 70 | 96 | 100 | 87 | 84 | 4% | $1.0 \times 10^5$ |
| 68 | 93 | 99 | 68 | 93 | 99 | 85 | 82 | 4% | $1.5 \times 10^5$ |
| 67 | 90 | 99 | 66 | 90 | 99 | 83 | 80 | 3% | $2.0 \times 10^5$ |
| 64 | 89 | 100 | 65 | 89 | 99 | 82 | 79 | 4% | $2.5 \times 10^5$ |
| 65 | 89 | 99 | 63 | 88 | 100 | 82 | 78 | 4% | $3.0 \times 10^5$ |
| 61 | 88 | 100 | 62 | 88 | 100 | 78 | 77 | 1% | $3.5 \times 10^5$ |
| 62 | 87 | 100 | 61 | 88 | 100 | 78 | 77 | 2% | $4.0 \times 10^5$ |
| 60 | 87 | 100 | 60 | 87 | 100 | 74 | 76 | 2% | $4.5 \times 10^5$ |
| 59 | 86 | 100 | 59 | 86 | 100 | 74 | 75 | 2% | $5.0 \times 10^5$ |
| 60 | 85 | 100 | 59 | 86 | 100 | 74 | 75 | 2% | $5.5 \times 10^5$ |
| 59 | 85 | 100 | 59 | 86 | 100 | 75 | 75 | 0% | $6.0 \times 10^5$ |
| 58 | 88 | 100 | 58 | 86 | 100 | 72 | 75 | 4% | $6.5 \times 10^5$ |
| 58 | 87 | 100 | 58 | 87 | 100 | 75 | 75 | 0% | $7.0 \times 10^5$ |
| 58 | 84 | 100 | 58 | 86 | 100 | 73 | 75 | 3% | $7.5 \times 10^5$ |
| 58 | 87 | 100 | 58 | 86 | 100 | 74 | 75 | 1% | $8.0 \times 10^5$ |
| 57 | 85 | 100 | 57 | 85 | 100 | 73 | 74 | 1% | $8.5 \times 10^5$ |
| 57 | 85 | 100 | 57 | 85 | 100 | 72 | 73 | 3% | $9.0 \times 10^5$ |
| 58 | 85 | 100 | 58 | 85 | 100 | 72 | 74 | 3% | $9.5 \times 10^5$ |
| 57 | 85 | 100 | 57 | 85 | 100 | 71 | 74 | 4% | $1.0 \times 10^6$ |

The predicted results of the actuation function index *A* for the cyclic load with stress amplitude of 24 MPa are compared with the corresponding experimental results, as shown in the third column block of Table 3. It can be found that the maximum relative error between the predicted result and our experimental result is 4%.

As mentioned before, the established model can be used to predict the actuation function degradation of the MFC actuator for an arbitrary cyclic load within the stress amplitude range of 12 MPa to 30 MPa. It is used to predict the degenerate curve of the MFC actuator undergoing the cyclic loading with stress amplitudes of 16 MPa, 20 MPa, 22 MPa, 26 MPa and 28 MPa, respectively. The predicted actuation function degradation curves are shown with the red solid line in Figure 11.

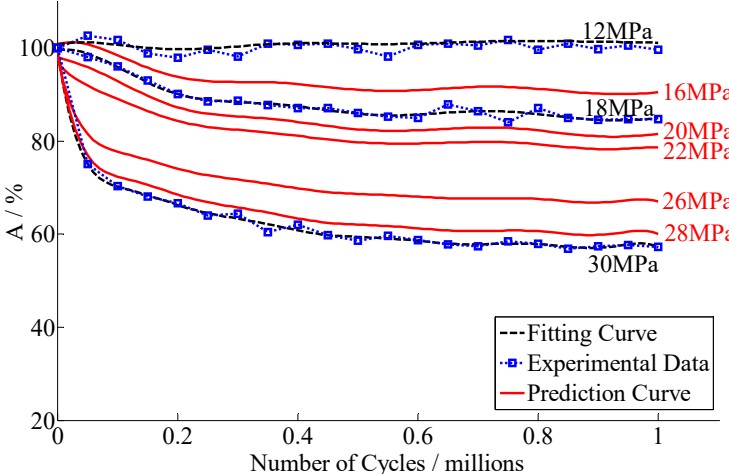

**Figure 11.** Predicted actuation function degradation curves (red solid line) of the MFC actuator undergoing cyclic load with stress amplitudes of 16 MPa, 20 MPa, 22 MPa 26 MPa and 28 MPa.

## 5. Conclusions

In the present study, the actuation function degradation degree of the MFC actuator undergoing cyclic loads was experimentally measured. Actuation function degradation experiments with four different stress amplitudes of cyclic loads have been implemented. Experimental results show that, when undergoing cyclic load with a stress amplitude of 12 MPa corresponding to a dynamic strain amplitude of the MFC is 400 $\mu\varepsilon$, no more than 10% degradation of piezoelectric actuation function has been observed. For a cyclic load with a stress amplitude of 18 MPa, 24 MPa and 30 MPa, the corresponding dynamic strain amplitudes of MFC are 600 $\mu\varepsilon$, 800 $\mu\varepsilon$ and 1000 $\mu\varepsilon$, respectively. The piezoelectric actuation function of the MFC actuator degenerates with the increase of loading cycles. Stress amplitude of the cyclic load is the key factor which determines the degenerate rate and degree of the MFC actuation function.

The higher the stress amplitude of the cyclic load is, the faster the degenerate rate will be. For certain loading cycles, the higher the stress amplitude of the cyclic load is, the greater degradation degree of the MFC actuation function will be.

Based on the measured data for the actuation function degradation of the MFC actuator undergoing cyclic load, an RBF neural network learning algorithm was adopted to establish a model for predicting the actuation function degradation of the MFC actuator undergoing arbitrary cyclic load within the linear stress limit range. Using the established actuation function degradation prediction model, the piezoelectric actuation function index *A* of the MFC actuator can be obtained for arbitrary cyclic load within the concerned stress amplitude limit for given cyclic loading cycles. Meanwhile, the actuation function degradation process of the MFC actuator undergoing arbitrary cyclic load within the concerned stress amplitude range can be graphically shown by using the degradation data provided by the established model. More importantly, with the help of this prediction model, it is possible to predict the function lifespan of the MFC actuator considering complex stress conditions in a practical service environment.

**Author Contributions:** Funding acquisition, W.W Investigation, W.W and Z.Z. Project administration, W.W. Writing—original draft, W.W. and Z.Z.; Writing—review & editing, W.W. and Z.Y.

**Funding:** This work was supported by the National Natural Science Foundation of China [Grant No. 11502208].

**Conflicts of Interest:** The authors declare no conflict of interest.

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
