# Peer review of "Experiment and Modeling on Macro Fiber Composite Stress-Induced Actuation Function Degradation"

_applsci, doi:10.3390/app9214714_

Round 1

Reviewer 1 Report

The problem presented in the work is very important and current. Piezoelectric actuators are widely used (vibration measurement, vibration reduction etc.). It is important to have a tool and a method to check their degradation and viability..
At work, the results are supported by the conducted experiment and constitute the basis for using such a solution.
Remark:
pages 4/5
table title1 please move from page 4 to page 5

Author Response

Dear Reviewer,

      Thank you for your kindly comments and suggestions.

Table title 1 has been moved from page 4 to page 5.

Thank you again for your review.

Best regards

Point 1:pages 4/5 table title1 please move from page 4 to page 5

Response 1: 

Table title 1 has been moved from page 4 to page 5.

Reviewer 2 Report

This manuscript investigates impacts of the fatigue life time on the MFC actuators. The authors experimentally have shown that the tip displacement of the MFC will be noticeably reduced after certain cyclic loads. In order to predict the performance degradation of the MFC for an arbitrary excitation load the neural network model presented. This work has some novel contributions, therefore, it can be considered for publication after the major revisions. The authors should address my following concern/comments:

First of all, the cited studies in Refs. 1-9 are quite old. It would be great if the authors provide the more recent studies. For instance: https://doi.org/10.3390/s19092196 https://doi.org/10.1109/ICSENS.2018.8589817 Since the MFC is placed on the steel beam, how the authors can ensure the performance degradation is only related to the MFC and no steel beam? It is not obvious why the authors focused on the tip displacement, why not the generated voltage? In presented the working flow chart in Fig. 1 what does “A stable” mean? Please clarify it. 9-11report the tip displacement degradation. I would recommend to the authors also report the frequency degradation (similar to utilized methodology for these figures) Why 25 hidden layers is selected? The authors can refer to this paper (https://doi.org/10.3390/proceedings2130930 ) and explain their own rotational.

Author Response

Dear Reviewer,

      Thank you for your comments and suggestions.

Best regards

Reviewer 3 Report

The paper is presented nicely. All figures and organization of the paper is good. However, little bit concerned about the impact of the work. How much this work will help the area. 

Author Response

(The authors gave the same response as above.)

Reviewer 4 Report

This study is focused on the experimental observation of actuation function degenerative behavior and actuation function degradation prediction of MFC actuator undergoing cyclic load. Based on the experimental results, authors adopt a radial basis function (RBF) neural network learning algorithm to establish a model, which can be used to predict the actuation function degradation degree of MFC actuator undergoing arbitrary stress amplitude of cyclic load within concerned stress amplitude range. In order this work to be accepted for publication, authors should clarify the following minor or major points:

1) Authors should clarify clearly the novelty of their work in their manuscript

2) The literature review is very poor and should be extended.

3) The language used in the manuscript needs improvement

4) Authors should describe what parameters At and A0 mean directly after the Eq. (1)

5) Since in Figure 3 presents the layout of the cyclic loading experiment for MFC actuator, the corresponding photo in Figure 4 seems needless

6) All parameters in the manuscript should be in italic form

7) Authors could use only one figure than six in Figure 12 showing the predicted curves for different loading stress amplitude comparing with the experimental results

8) Authors should discuss about the generality of the predictive model 

9) Authors use the radial basis function (RBF) neural network learning algorithm. Authors should use some references  and a brief description about this algorithm.

Author Response

(The authors gave the same response as above.)

Round 2

Reviewer 2 Report

The authors have addressed all my comments, then the current version of the manuscript can be considered for publication. 

Reviewer 4 Report

This work could be published in its present form.